# Changed Amino Acids in NAFLD and Liver Fibrosis: A Large Cross-Sectional Study without Influence of Insulin Resistance

**DOI:** 10.3390/nu12051450

**Published:** 2020-05-17

**Authors:** Takuma Hasegawa, Chikara Iino, Tetsu Endo, Kenichiro Mikami, Masayo Kimura, Naoya Sawada, Shigeyuki Nakaji, Shinsaku Fukuda

**Affiliations:** 1Department of Gastroenterology, Hirosaki University Graduate School of Medicine, Hirosaki 036-8562, Japan; tue4c_hasegawa@yahoo.co.jp (T.H.); endo16@hirosaki-u.ac.jp (T.E.); ms.is.twoday@gmail.com (M.K.); sawada226@gmail.com (N.S.); sfukuda@hirosaki-u.ac.jp (S.F.); 2Department of Internal Medicine, Owani Hospital, Owani, 5-3 Hagurokan Owani, Aomori 038-0292, Japan; kmikami@hirosaki-u.ac.jp; 3Department of Social Medicine, Hirosaki University Graduate School of Medicine, Hirosaki 036-8562, Japan; nakaji@hirosaki-u.ac.jp

**Keywords:** nonalcoholic fatty liver disease, nonalcoholic steatohepatitis, liver fibrosis, amino acids, insulin resistance

## Abstract

Altered amino acid levels have been found in nonalcoholic fatty liver disease (NAFLD) and nonalcoholic steatohepatitis (NASH). However, it is not clear whether this alteration is due to altered hepatic metabolism or insulin resistance. The aim of this study was to clarify the association among amino acid levels, fatty liver, and liver fibrosis while eliminating the influence of insulin resistance. NAFLD and liver fibrosis were diagnosed using transient elastography and subjects were divided into three groups: normal, NAFLD, and liver fibrosis. To exclude the influence of insulin resistance, the subjects were matched using the homeostasis model assessment of insulin resistance (HOMA-IR). The amino acid serum levels were compared among the groups. Of 731 enrolled subjects, 251 and 33 were diagnosed with NAFLD and liver fibrosis. Although significant differences were observed among the groups in the serum levels of most amino acids, all but those of glutamate and glycine disappeared after matching for HOMA-IR. The multivariate logistic regression revealed that glutamate, glycine, and HOMA-IR were independent risk factors for liver fibrosis. The altered serum levels of most amino acids were associated with insulin resistance, while the increase in glutamate and the decrease in glycine levels were strongly associated not only with insulin resistance, but also with altered liver metabolism in patients with liver fibrosis.

## 1. Introduction

Nonalcoholic fatty liver disease (NAFLD) is one of the most common causes of chronic liver disease worldwide [1]. The clinical/histological type of NAFLD ranges from simple steatosis to nonalcoholic steatohepatitis (NASH), fibrosis, and cirrhosis. NAFLD is currently recognized as the hepatic representation of metabolic syndrome and its pathogenesis has been associated with insulin resistance and altered hepatic metabolism [2,3].

Serum levels of amino acids, in particular, branched-chain amino acids (BCAA) and glutamate, have been found to be increased in type 2 diabetes mellitus and have been implicated in insulin resistance [4,5,6,7]. Although the underlying mechanisms are not fully understood, amino acids may contribute to development of obesity-associated insulin resistance [4,7]. Several recent studies revealed altered amino acid levels in NAFLD and NASH as well [7,8,9,10]. Moreover, the serum levels of amino acids have been associated with the pathogenesis of NAFLD and its progression to NASH [8]. Serum amino acid levels may be elevated in obesity-associated NAFLD, and mitochondrial dysfunction in NAFLD may result in impaired amino acids metabolism [7]. However, their role has not been elucidated; thus, it remains unclear whether the change in the serum amino acid levels in subjects with NAFLD and NASH is the result of altered hepatic metabolism or insulin resistance. Furthermore, in most of these studies, the sample size was small [8,9,10]. 

The aim of this study was to clarify the association among amino acid levels, fatty liver, and liver fibrosis while eliminating the influence of insulin resistance. Therefore, we investigated the amino acid serum levels in a large cohort and compared them among normal subjects, subjects with NAFLD, and those with liver fibrosis by matching the degree of insulin resistance.

## 2. Materials and Methods

### 2.1. Study Subjects

In total, 1056 healthy subjects who participated in the Iwaki Health Promotion Project Health Survey held in June 2018 in Aomori prefecture, northern Japan (Figure 1) were invited to participate in this study. All subjects were explained the details of the examination and the principal aims of the study, and only those who provided written informed consent were included. The following were the exclusion criteria: (1) positive hepatitis B surface antigen or anti-hepatitis C virus test; (2) excessive alcohol intake (men, > 30 g/day; women, > 20 g/day); and (3) history of use of medications associated with steatosis, such as amiodarone, methotrexate, prednisolone, and tamoxifen. 

### 2.2. Transient Elastography

Transient elastography with measurement of liver stiffness (LS) and controlled attenuation parameter (CAP), which quantifies hepatic fat, was performed using FibroScan (Echosens, Paris, France). Examinations were performed by five well-trained hepatology specialists. In cases when LS or CAP values could not be obtained after at least 10 attempts, the examinations were considered to have failed. LS measurement was considered unreliable if the ratio of its interquartile range with the median LS value was greater than 0.30, with the median LS value set at 7.1 kPa, according to previously established criteria [11]. 

NAFLD was diagnosed when the CAP was ≥ 237.8 dB [12], and liver fibrosis was diagnosed when the LS was ≥ 7 kPa and the CAP was ≥ 237.8 dB, according to previously established criteria [13]. Based on the transient elastography findings, the subjects were divided into three groups: normal, NAFLD, and liver fibrosis group. 

### 2.3. Serum Amino Acid Profiles

After an overnight fast, blood samples were collected from all subjects on the morning of the transient elastography examination. The serum amino acid profiles were evaluated by high-performance liquid chromatography at LSI Medience Corporation (Tokyo, Japan). Briefly, sulfosalicylic acid was added to plasma to a final concentration of 5%. The samples were then incubated on ice for 15 min and centrifuged to remove precipitated proteins. The extracts were then analyzed for amino acid content using an amino acid analyzer [14]. The serum amino acid profiles were compared among three groups.

### 2.4. Clinical Parameters

The following clinical parameters were recorded on the same day as the transient elastography examination: sex, age, height, body weight, body mass index (calculated by dividing the weight in kilograms by the squared height in meters), results for hepatitis B surface antigen or anti-hepatitis C virus test, and the levels of amino acids, aspartate aminotransferase, alanine aminotransferase, gamma-glutamyl transpeptidase, albumin, total bilirubin, glucose, insulin, hemoglobin A1c (HbA1c), uric acid, total cholesterol, high-density lipoprotein cholesterol, low-density lipoprotein cholesterol, triglycerides, and platelets. Plasma concentrations of glucose, insulin, liver enzymes, and lipids were measured as reported [15]. The intake of total energy and protein was calculated based on the results of the brief self-administered diet history questionnaire (BDHQ), a convenient diet assessment questionnaire developed in Japan [16]. The BDHQ included questions concerning the intake frequency of 58 food and beverage items commonly consumed in Japan.

### 2.5. Insulin Resistance and Diabetes

The insulin resistance index was calculated using the homeostasis model assessment of insulin resistance (HOMA-IR), as follows: fasting glucose (mg/dL) × fasting insulin (μU/mL)/405 [17]. We investigated the correlation between the HOMA-IR index and the serum amino acid levels. To exclude the influence of insulin resistance, the subjects of the three groups were matched using calipers of a width equal to an HOMA-IR index of 0.12. Type 2 diabetes was defined as a fasting plasma glucose ≥ 126 mg/dl and HbA1c ≥ 6.5%, or treatment for diabetes.

### 2.6. Statistical Analysis

Statistical analysis of the clinical data was performed using JMP ver. 12.1 (SAS Institute, Cary, NC, USA) and R software (R Foundation for Statistical Computing, version R-3.4.3). Categorical variables were shown as frequencies and percentages, while continuous variables were shown as means with standard deviation or medians with interquartile ranges. Categorical variables were compared using the chi-square test with Holm p-adjustment, and continuous variables were compared using the parametric Dunnett test and nonparametric Steel test. Spearman’s rank correlation coefficients were calculated to determine the correlation between the serum amino acid levels and the HOMA-IR index. A *p* value less than 0.05 was considered statistically significant. We used logistic regression analysis to identify the variable independently associated with liver fibrosis among the HOMA-IR index and the serum levels of the amino acids that had a significant association with liver fibrosis in the comparison analysis between normal and subjects with liver fibrosis.

### 2.7. Statement of Human Rights

All procedures followed were in accordance with the ethical standards of the institutional and/or national research committee (Hirosaki University Medical Ethics Committee; Authorization number: 2018-062) and with the Helsinki declaration of 1975, as revised in 2008.

### 2.8. Informed Consent

Informed consent was obtained from all individual participants included in the study.

## 3. Results

### 3.1. Participants’ Characteristics and Amino Acid Compositions

Of the 1056 subjects who participated in the health survey, 797 were eligible for inclusion in the study and underwent transient elastography. Of these, 35 subjects in whom the examination failed and 31 subjects with unreliable data were excluded (Figure 1). Finally, 731 subjects (33.7% males) with a mean age of 51.6 (range, 20–88) years who had reliable data were included in the analysis.

Among all subjects, 251 (35.6%) were diagnosed with NAFLD and 33 (4.7%) were diagnosed with liver fibrosis. The clinical parameters of the study subjects are summarized in Table 1. 

There were significant differences among the groups in most values, including the HOMA-IR index. In the intake of total energy and protein, there were significant differences among the groups. Significant differences among the groups were also observed in the serum levels of most amino acids (Table 2).

The principal coordinate analysis revealed significant differences in the amino acid composition among the groups (NAFLD vs. normal *p* < 0.05; and liver fibrosis vs. normal *p* < 0.05) (Figure 2).

### 3.2. Correlation of Serum Amino Acid Levels and HOMA-IR Index

In the analysis of the correlation between the amino acids and the HOMA-IR index, significant correlations were found for many of them. In particular, glutamate had the strongest correlation with the HOMA-IR index (Table 3).

### 3.3. Participants’ Characteristics and Amino Acid Compositions after Matching

After matching for HOMA-IR, there were 15 matched subjects in the three groups, and no significant differences were found in terms of the levels of glucose, insulin, and HbA1C, or the HOMA-IR index (Table 4). 

However, the serum levels of glutamate were significantly higher and those of glycine were significantly lower in subjects with liver fibrosis compared to the levels in normal subjects (Table 5). Moreover, the serum levels of alanine in subjects with NAFLD were significantly higher than those in normal subjects.

### 3.4. Risk Factors for Liver Fibrosis

The multivariate logistic regression analysis revealed that glutamate (OR: 1.03, *p* < 0.001), glycine (OR: 0.99, *p* = 0.041), and the HOMA-IR index (OR: 1.43, *p* < 0.001) were independently associated with liver fibrosis (Table 6). 

## 4. Discussion

In this study, by principal coordinate analysis, we determined significant differences in the serum levels and compositions of most amino acids among normal subjects, subjects with NAFLD, and those with liver fibrosis. Moreover, although a correlation was observed between insulin resistance and the serum levels of most amino acids, the significant differences for most amino acids disappeared after matching for insulin resistance. Therefore, insulin resistance would be one of the major factors that caused the alteration in the serum amino acids levels in NAFLD and liver fibrosis. However, after excluding the influence of insulin resistance, only a significant increase in serum glutamate levels and a significant decrease in serum glycine levels were observed in subjects with liver fibrosis. Moreover, the multivariate analysis also showed that glutamate and glycine, independently from insulin resistance, had a significant pathophysiological role in the development of liver fibrosis. Therefore, the serum levels of glutamate and glycine were associated not only with insulin resistance, but also with altered hepatic metabolism in subjects with liver fibrosis. 

Our results reveal increased levels of BCAAs and other amino acids, such as glutamate, glutamine, and alanine, and decreased levels of glycine in subjects with liver fibrosis. These findings are similar to those of previous studies on metabolic disease, including NAFLD [7,18]. Patients with NAFLD, among whom there is a high prevalence of metabolic disorders, including type 2 diabetes, have increased insulin resistance [19,20]. A recent study that investigated only BCAAs revealed that NAFLD and elevated BCAA levels could have a synergistic effect on the development of type 2 diabetes and pointed out that insulin resistance and mitochondrial dysfunction were important mechanisms for the association between BCAA and NAFLD [7]. The authors suggested that the altered amino acid concentration was associated with increased insulin resistance. Indeed, in the current study, the serum levels of most amino acids, including glutamate and glycine, were correlated with insulin resistance, which is in line with the findings of previous studies [21,22]. 

In terms of liver fibrosis, we found a significant increase in the serum glutamate levels and a significant decrease in the serum glycine levels in subjects with liver fibrosis after matching by insulin resistance. Moreover, the results of the multivariate analysis also demonstrate that the serum levels of glutamate and glycine were independent factors for liver fibrosis. A previous longitudinal study of patients with NAFLD revealed that liver fibrosis was the most important liver histological feature independently associated with long-term overall mortality [23]. Therefore, patients with NAFLD should be carefully evaluated and monitored for liver fibrosis. A recent study that included evaluation of liver histology demonstrated that glutamate was the amino acid that was more strongly associated with the severity of fibrosis [10]. Glutamate and glycine are involved in glutathione synthesis. Glutathione is an important antioxidant molecule that directly reacts with reactive oxygen species (ROS) in conditions of oxidative stress. Mitochondrial dysfunction leads to the production of ROS, which is one of the key mechanisms accountable for the pathogenesis of NASH [24,25]. The overproduction of ROS contributes to depletion of glutathione, which stimulates glutathione synthesis [26,27,28]. This leads to increased consumption of glutamate and glycine, which are involved in glutathione synthesis. Glutamate is also produced by transamination of alanine and aspartate. The increase in glutamate levels would be attributed to the activity of γ-glutamyl transpeptidase, which promotes glutamate release into the blood during the transamination of glutathione [10]. Regarding the levels of glycine, which is consumed in glutathione synthesis, a study on the metabolism of glutathione revealed that glycine was the limiting substrate for glutathione synthesis [9]. Therefore, glycine would be decreased in case of stimulation of glutathione synthesis. Considering the above, the serum levels of glutamate and glycine would be strongly associated with liver fibrosis beyond the influence of insulin resistance.

The strengths of this study include the large number of participants and the availability of extensive data. However, there are also some limitations to this study. First, the diagnosis of NAFLD and liver fibrosis in the present study was made using transient elastography. Liver biopsy remains the gold standard for diagnosis of NAFLD and liver fibrosis; however, it is an invasive procedure. Performing liver biopsy in the present study based on a mass survey would have been unethical. Although the cutoff value for CAP and LS remains controversial, transient elastography is considered to be an effective noninvasive alternative to evaluate hepatic steatosis and liver fibrosis in patients with chronic liver disease [29,30]. The American Association for the Study of Liver Diseases guidelines state that transient elastography is a clinically useful tool for identifying advanced fibrosis in patients with NAFLD [31]. The joint European Association guidelines on NAFLD state that transient elastography is an acceptable non-invasive procedure for identification of patients at low risk of advanced fibrosis or cirrhosis [32]. Second, after matching, we evaluated three groups with high levels of insulin. In normal subjects, serum levels of insulin after matching were much higher than those before matching because of matching for insulin resistance. It is a limitation of the matching. Third, the primary limitation of this study was the limited number of fibrosis subjects. However, the sample size used here is almost the same as in previous studies [8,9,10].

## 5. Conclusions

We found altered amino acid serum levels in subjects with NAFLD and liver fibrosis. The changes in the serum levels of most amino acids were associated with insulin resistance, while the increase in glutamate and the decrease in glycine levels were strongly associated not only with insulin resistance, but also with altered liver metabolism in patients with liver fibrosis.

## Figures and Tables

**Figure 1 nutrients-12-01450-f001:**
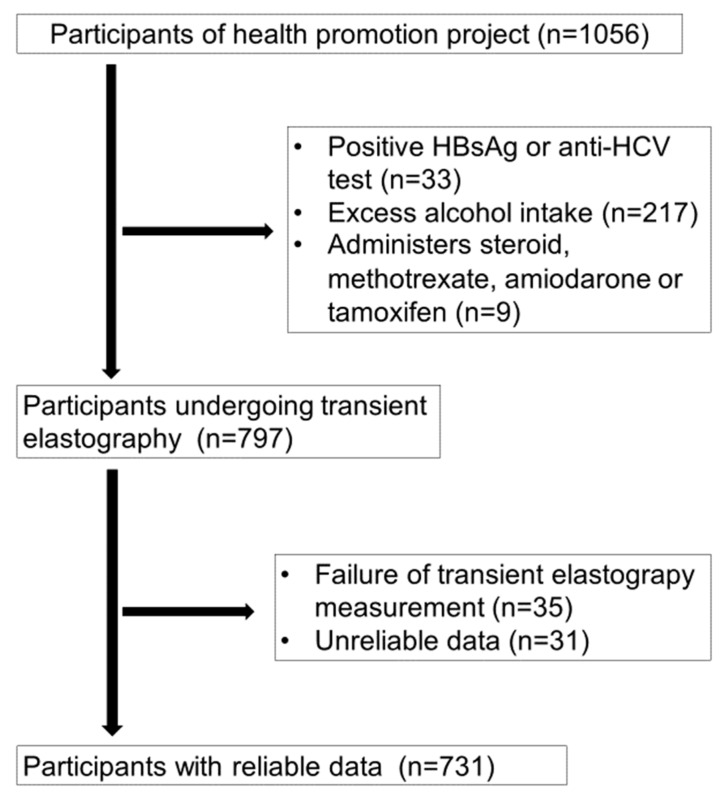
Study enrollment flow chart.

**Figure 2 nutrients-12-01450-f002:**
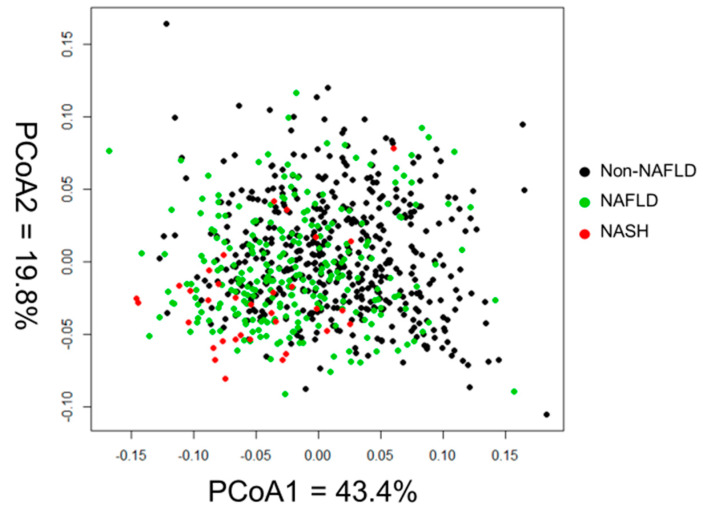
Principal coordinate analysis of the amino acid composition among normal subjects, subjects with nonalcoholic fatty liver diseases, and subjects with liver fibrosis.

**Table 1 nutrients-12-01450-t001:** Participants’ characteristics at baseline.

	Normal	NAFLD	Fibrosis	*p* Value	*p* Value
				Normal vs NAFLD	Normal vs Fibrosis
n	420	251	33		
Age (years)	49.6 ± 16.2	54.5 ± 14.9	54.1 ± 14.4	<0.001	0.210
Gender, Male	132 (29.8%)	100 (39.8%)	15 (45.5%)	0.017	0.111
BMI (kg/m^2^)	21.3 ± 2.7	24.3 ± 3.3	28.8 ± 5.0	<0.001	<0.001
Glucose (mg/dL)	91 ± 9	97 ± 14	124 ± 36	<0.001	<0.001
Insulin (µU/mL)	4.7 ± 1.9	7.3 ± 8.9	11.4 ± 6.1	<0.001	<0.001
HOMA-IR	1.1 ± 0.5	1.8 ± 1.9	3.6 ± 2.6	<0.001	<0.001
Total bilirubin (mg/dL)	0.9 ± 0.3	0.8 ± 0.3	0.8 ± 0.3	0.010	0.376
AST (U/L)	20.6 ± 5.8	23.0 ± 11.4	35.8 ± 21.0	0.002	<0.001
ALT (U/L)	17.4 ± 8.0	25.8 ± 27.8	50.7 ± 42.3	<0.001	<0.001
GGT (U/L)	23.1 ± 15.9	31.9 ± 29.3	57.2 ± 62.4	<0.001	<0.001
Total cholesterol (mg/dL)	198 ± 34	208 ± 38	209 ± 25	<0.001	0.163
Triglycerides (mg/dL)	76 ± 38	103 ± 58	136 ± 52	<0.001	<0.001
HDL cholesterol (mg/dL)	68 ± 16	61 ± 16	54 ± 21	<0.001	<0.001
LDL cholesterol (mg/dL)	113 ± 28	126 ± 32	127 ± 23	<0.001	0.018
Albumin (g/dL)	4.4 ± 0.3	4.4 ± 0.3	4.4 ± 0.3	0.310	0.920
Platelets (×10⁴/µL)	26.3 ± 5.9	27.1 ± 6.1	25.4 ± 5.2	0.180	0.611
HbA1c (%)	5.7 ± 0.4	5.8 ± 0.5	7.0 ± 1.6	<0.001	<0.001
Type 2 diabetes	10 (2.4%)	15 (6.0%)	13 (39.4%)	0.034	<0.001
Total energy intake (kcal/day)	1807 ± 547	1809 ± 548	1796 ± 569	0.998	0.991
Protein intake (g/day)	71 ± 26	73 ± 27	67 ± 28	0.705	0.622
CAP (dB/m)	183 ± 36	277 ± 32	320 ± 31	<0.001	<0.001
LS (kPa)	4.2 ± 1.1	4.4 ± 1.1	9.8 ± 2.7	0.175	<0.001

Data are presented as means ± standard deviation. ALT, alanine aminotransferase; AST, aspartate aminotransferase; BMI, body mass index; CAP, controlled Attenuation Parameter; GGT, γ-glutamyl transpeptidase; HbA1c, hemoglobin A1c; HDL, high density lipoprotein; HOMA-IR, homeostasis model assessment of insulin resistance; LDL, low density lipoprotein; LS, liver stiffness; NAFLD, nonalcoholic fatty liver disease.

**Table 2 nutrients-12-01450-t002:** Amino acid composition at baseline.

	Normal	NAFLD	Fibrosis	*p* Value	*p* Value
				Normal vs. NAFLD	Normal vs. Fibrosis
Amino acid (nmol/mL)	n = 420	n = 251	n = 33		
Valine	199 (178–224)	223 (197–245)	257 (222–284)	<0.001	<0.001
Leucine	102 (92–118)	115 (100–130)	132 (112–150)	<0.001	<0.001
Isoleucine	52 (46–61)	58 (49–68)	69 (57–78)	<0.001	<0.001
Methionine	24 (22–27)	25 (22–27)	24 (21–27)	0.111	0.781
Lysine	194 (174–211)	198 (180–221)	207 (188–226)	0.010	0.010
Phenylalanine	54 (50–59)	56 (52–62)	58 (53–66)	<0.001	0.003
Histidine	79 (73–84)	80 (75–85)	78 (73–82)	0.020	0.712
Threonine	123 (107–143)	125 (109–146)	117 (110–139)	0.474	0.924
Tryptophan	44 (40–49)	46 (41–51)	45 (39–53)	0.007	0.417
Tyrosine	55 (49–63)	61 (54–68)	71 (62–78)	<0.001	<0.001
Glycine	241 (211–284)	223 (196–271)	192 (165–216)	<0.001	<0.001
Alanine	310 (261–369)	356 (307–402)	387 (347–419)	<0.001	<0.001
Serine	120 (108–138)	115 (102–130)	117 (100–128)	0.002	0.077
Arginine	80 (68–93)	83 (71–94)	82 (74–92)	0.214	0.715
Cystine	27 (20–34)	31 (25–37)	34 (28–41)	<0.001	<0.001
Asparagine	52 (47–57)	50 (45–55)	48 (45–52)	0.040	0.035
Glutamine	590 (543–637)	593 (546–637)	575 (519–613)	0.953	0.157
Proline	129 (107–157)	141 (118–174)	152 (127–166)	<0.001	0.002
Aspartic acid	4 (3–5)	4 (3–5)	5 (4–5)	<0.001	<0.001
Glutamate	42 (32–53)	53 (39–67)	69 (57–85)	<0.001	<0.001

Data are presented as medians with interquartile ranges. NAFLD, nonalcoholic fatty liver disease.

**Table 3 nutrients-12-01450-t003:** Correlation of serum amino acid levels and HOMA-IR index.

	r	*p* Value
Valine	0.359	<0.001
Leucine	0.253	<0.001
Isoleucine	0.274	<0.001
Methionine	−0.014	0.697
Lysine	0.046	0.214
Phenylalanine	0.164	<0.001
Histidine	−0.080	0.030
Threonine	−0.033	0.370
Tryptophan	0.112	0.002
Tyrosine	0.274	<0.001
Glycine	−0.227	<0.001
Alanine	0.337	<0.001
Serine	−0.170	<0.001
Arginine	0.004	0.922
Cystine	0.241	<0.001
Asparagine	−0.250	<0.001
Glutamine	−0.142	<0.001
Proline	0.225	<0.001
Aspartic acid	0.318	<0.001
Glutamate	0.494	<0.001

HOMA-IR, homeostasis model assessment of insulin resistance.

**Table 4 nutrients-12-01450-t004:** Participants’ characteristics after matching for insulin resistance.

	Normal	NAFLD	Fibrosis	*p* Value	*p* Value
				Normal vs NAFLD	Normal vs Fibrosis
n	15	15	15		
Age (years)	55.8 ± 14.4	53.2 ± 14.4	57.4 ± 13.7	0.843	0.937
Gender, Male	6 (40.0%)	3 (20.0%)	6 (40.0%)	0.696	0.999
BMI (kg/m^2^)	22.0 ± 3.4	26.9 ± 4.7	26.6 ± 4.3	0.007	0.011
Glucose (mg/dL)	104 ± 16	102 ± 20	112 ± 31	0.989	0.544
Insulin (µU/mL)	8.1 ± 2.6	8.3 ± 2.4	7.7 ± 2.5	0.981	0.884
HOMA-IR	2.2 ± 1.0	2.2 ± 0.9	2.2 ± 0.9	0.999	0.999
Total bilirubin (mg/dL)	0.9 ± 0.3	0.7 ± 0.2	0.9 ± 0.3	0.134	0.852
AST (U/L)	20.9 ± 5.3	25.7 ± 8.2	39.9 ± 26.4	0.645	0.007
ALT (U/L)	20.2 ± 6.9	30.0 ± 13.3	58.1 ± 58.6	0.680	0.012
GGT (U/L)	27.9 ± 14	36.8 ± 15.9	54.0 ± 33.5	0.487	0.008
Total cholesterol (mg/dL)	203 ± 29	210 ± 37	211 ± 20	0.778	0.712
Triglycerides (mg/dL)	90 ± 41	139 ± 70	113 ± 33	0.026	0.366
HDL cholesterol (mg/dL)	66 ± 16	56 ± 14	56 ± 13	0.127	0.156
LDL cholesterol (mg/dL)	117 ± 26	127 ± 28	129 ± 17	0.444	0.286
Albumin (g/dL)	4.4 ± 0.3	4.3 ± 0.3	4.4 ± 0.3	0.419	0.827
Platelets (×10⁴/µL)	28.3 ± 7.7	27.0 ± 5.3	25.8 ± 4.8	0.805	0.466
HbA1c (%)	6.1 ± 0.8	6.1 ± 0.8	6.5 ± 1.4	0.983	0.554
Type 2 diabetes	0	0	7 (46.7%)	0.999	0.008
CAP (dB/m)	186 ± 39	280 ± 39	306 ± 28	<0.001	<0.001
LS (kPa)	4.8 ± 1.3	4.4 ± 1.2	9.8 ± 1.9	0.754	<0.001

Data are presented as means ± standard deviation. ALT, alanine aminotransferase; AST, aspartate aminotransferase; BMI, body mass index; CAP, controlled Attenuation Parameter; GGT, γ-glutamyl transpeptidase; HbA1c, hemoglobin A1c; HDL, high density lipoprotein; HOMA-IR, homeostasis model assessment of insulin resistance; LDL, low density lipoprotein; LS, liver stiffness; NAFLD, nonalcoholic fatty liver disease.

**Table 5 nutrients-12-01450-t005:** Amino acid composition after matching for insulin resistance.

	Normal	NAFLD	Fibrosis	*p* Value	*p* Value
				Normal vs NAFLD	Normal vs Fibrosis
Amino acid (nmol/mL)	n = 15	n = 15	n = 15		
Valine	211 (177–241)	242 (220–257)	233 (212–271)	0.081	0.117
Leucine	102 (94–130)	118 (108–129)	117 (110–143)	0.208	0.151
Isoleucine	52 (45–64)	59 (53–63)	63(54–74)	0.299	0.145
Methionine	23 (21–27)	25 (23–28)	24 (22–25)	0.320	0.904
Lysine	186 (169–214)	194 (181–222)	204 (193–230)	0.546	0.178
Phenylalanine	56 (52–60)	62 (55–64)	56 (53–65)	0.164	0.717
Histidine	76 (70–81)	81 (79–85)	80 (78–87)	0.057	0.178
Threonine	126 (114–135)	135 (124–153)	125 (114–140)	0.260	0.992
Tryptophan	41 (39–47)	48 (43–51)	43 (38–50)	0.127	0.872
Tyrosine	59 (53–67)	67 (62–70)	66 (60–75)	0.138	0.260
Glycine	260 (229–284)	210 (183–295)	192 (173–229)	0.321	0.016
Alanine	332 (310–357)	369 (362–475)	360 (321–424)	0.004	0.242
Serine	120 (110–131)	109 (99–121)	108 (100–129)	0.260	0.401
Arginine	86 (66–92)	84 (76–91)	82 (73–85)	0.999	0.992
Cystine	31(25–42)	34 (26–40)	33 (28–41)	0.985	0.998
Asparagine	51(49–52)	51 (48–55)	49 (47–53)	0.824	0.969
Glutamine	618(586–647)	593 (540–644)	610 (538–631)	0.731	0.389
Proline	138(114–177)	147 (124–172)	148 (123–156)	0.997	0.999
Aspartic acid	4 (3–5)	4 (3–4)	4 (4–5)	0.913	0.771
Glutamate	50 (34–58)	57 (42–65)	59 (56–67)	0.574	0.045

Data are presented as medians with interquartile ranges. NAFLD, nonalcoholic fatty liver disease.

**Table 6 nutrients-12-01450-t006:** Multivariate analysis of risk factors for liver fibrosis.

Variables	Multivariable		
	OR	95% CI	*p* value
HOMA-IR	1.43	1.12–1.91	<0.001
Glutamate	1.03	1.01–1.05	<0.001
Glycine	0.99	0.98–0.99	0.041

HOMA-IR, homeostasis model assessment of insulin resistance; OR, odds ratio; CI, confidence interval.

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
