# Peer review of "Changed Amino Acids in NAFLD and Liver Fibrosis: A Large Cross-Sectional Study without Influence of Insulin Resistance"

_nutrients, 2020, doi:10.3390/nu12051450_

Round 1

Reviewer 1 Report

This study is very interesting and important for understanding the profile of amino acids for liver fibrosis patients. From the results view, the alterations of amino acids composition are found only few amino acids, the question is the amino acids specially glutamate and glycine is not good diagnosis biomarker for the determined extent of liver fibrosis. I guess some of liver fibrosis patients could be malnutrition or nutrient

deficiency. How about the evaluation of essential amino acid is more reliable? The language needs to be edited by native English person.

The title is hard to learn the detail of real outcome in this study, this is a Cohort study????

by gas chromatography-mass spectrometry at LSI Medience Corporation (Tokyo, Japan). GC-MS is employed to analyze amino acids. Why is not LC-MS??

IRB number should be provided in M&M section

From Table 1 list, how many peoples or percentage are diagnosed type 2 diabetes, suggesting diabetes or insulin resistance should be included as a negative control.

Compared Insulin level in table 1 about 4.7 μU/mL and 8.1 μU/mL in table 4. Why 15 normal subjects showed such high level but most normal peoples showed low level??? almost 2 fold difference why??

Author Response

This study is very interesting and important for understanding the profile of amino acids for liver fibrosis patients. From the results view, the alterations of amino acids composition are found only few amino acids, the question is the amino acids specially glutamate and glycine is not good diagnosis biomarker for the determined extent of liver fibrosis. I guess some of liver fibrosis patients could be malnutrition or nutrient deficiency. How about the evaluation of essential amino acid is more reliable? The language needs to be edited by native English person.

Reply: You raise an important question. We also think some of liver fibrosis patients could be malnutrition or nutrient deficiency. Therefore, we investigated the intake of total energy and protein by brief self-administered diet history questionnaire (BDHQ). Although the intake of total energy and protein in subjects with fibrosis were lower than those in normal subjects, there were not significant differences between them. The reason would be that this study subjects are heathy. We added total energy intake and protein intake by BDHQ in Table1 and Results (P5, Line 134).

The paper has been edited and rewritten by an experienced scientific editor, who has improved the grammar and stylistic expression of the paper. Moreover, the additional sentences have been reedited and rewritten.

The title is hard to learn the detail of real outcome in this study, this is a Cohort study????

Reply: As reviewer’s comment, we changed the title. This study is cross-sectional study. Therefore, we added the words “cross-sectional study” in the title (P1, Line 3).

by gas chromatography-mass spectrometry at LSI Medience Corporation (Tokyo, Japan). GC-MS is employed to analyze amino acids. Why is not LC-MS??

Reply: As reviewer’s comment, we mistook the method for the assay of serum amino acid. The method was high-perfomance liquid chromatography. We changed the sentences in methods and added the reference and some details for the assay of serum amino acid (P2, Lines 73-77).

RB number should be provided in M&M section

Reply: Accordingly, we added the IRB number in 2.7. Statement of Human Rights (P3, Lines 112-113).

From Table 1 list, how many peoples or percentage are diagnosed type 2 diabetes, suggesting diabetes or insulin resistance should be included as a negative control.

Reply: As reviewer’s comment, we added the number and percentage of type 2 diabetes in Tale1 and Table4.

Compared Insulin level in table 1 about 4.7 μU/mL and 8.1 μU/mL in table 4. Why 15 normal subjects showed such high level but most normal peoples showed low level??? almost 2 fold difference why??

Reply: As reviewer’s comment, in normal subjects, serum level of insulin after matching was much higher than those before matching because of matching for insulin resistance. It is a limitation of matching. We added the sentences about these limitations into the limitation section (P10, Lines 232-234).

Reviewer 2 Report

The study is not well designed. Overall the novelty of the study is low and there are not enough discussion to fully support the claims. Therefore, it requires extensive re-interpretation and additional details/experiments before it can be properly evaluated.

Abstract

(1) There is no full name for all the abbreviation.

Introduction

(1) The authors should add more details about the association between amino acids and metabolic diseases such as diabetes and NAFLD by previous studies.

(2) “Furthermore, in most of these studies, the sample size was small.” Cite related literature for the statement.

Results

(1) Table 2 and 5: What is the unit for these amino acids?

(2) Figure 2: Principal coordinate analysis or Principal component analysis?

Discussion

(1) The authors should give more mechanism suspect for the association between BCAAs and NAFLD.

(2) The authors should point out the limited number of fibrosis cases as one limitation of this study.

Author Response

Abstract

There is no full name for all the abbreviation.

Reply: Accordingly, we added the full name for all the abbreviation (P1, Line 11), (P1, Line 12), (P1, Lines 17-18).

Introduction

The authors should add more details about the association between amino acids and metabolic diseases such as diabetes and NAFLD by previous studies.

Reply: As reviewer’s comment, we added more details about the association between amino acids and metabolic disease by previous studies (P1, Lines 37-38), (P1, Lines 41-42).

“Furthermore, in most of these studies, the sample size was small.” Cite related literature for the statement.

Reply: As reviewer’s comment, we cited related literature for the statement (P2, Line 45).

Results

Table 2 and 5: What is the unit for these amino acids?

Reply: As reviewer’s comment, we added the unit for amino acids in Table2 and 5.

Figure 2: Principal coordinate analysis or Principal component analysis?

Reply: Accordingly, we changed Principal component analysis to Principal coordinate analysis (P6, Line 142).

Discussion

The authors should give more mechanism suspect for the association between BCAAs and NAFLD.

Reply: As reviewer’s comment, insulin resistance and mitochondrial dysfunction were important mechanisms for the association between BCAA and NAFLD. We added these sentences in the Discussion section (P9, Lines 194-195).

The authors should point out the limited number of fibrosis cases as one limitation of this study.

Reply: As reviewer’s comment, the limitation of this study was the limited number of fibrosis subjects. We added the sentences about this limitation in the limitation section (P10, Lines 234-236).

Reviewer 3 Report

The current study investigated the association among amino acid levels, fatty liver, and liver fibrosis while eliminating the influence of insulin resistance. The study is of a high importance in the field of nonalcoholic fatty liver disease. After revision, it should be reconsidered for publication in Nutrients.

  1. Keywords: Please delete some of them, and keep 3-5 keywords.
  2. The item 2.2, there should be a space between “237.8” and “dB”.
  3. Please add the reference or some details for the assay of serum amino acid.
  4. Please add methods for the item 2.4.
  5. For the part of Results, please add several subtitles to show the data.
  6. Table 2, it is unclear about the data. Can you give the unit for each amino acid?

Author Response

Keywords: Please delete some of them, and keep 3-5 keywords.

Reply: Accordingly, we deleted 5 keywords.

The item 2.2, there should be a space between “237.8” and “dB”.

Reply: Accordingly, we added a space between“237.8” and “dB”.

Please add the reference or some details for the assay of serum amino acid.

Reply: As reviewer’s comment, we changed the sentences in methods and added the reference and some details for the assay of serum amino acid (P2, Lines 73-77).

Please add methods for the item 2.4.

Reply: As reviewer’s comment, we added the sentences about methods in the item 2.4 (P2, Lines 86-87).

For the part of Results, please add several subtitles to show the data.

Reply: As reviewer’s comment, we added the 4 subtitles in the part of Results.

Table 2, it is unclear about the data. Can you give the unit for each amino acid?

Reply: As reviewer’s comment, we added the unit for amino acids in Table2 and 5.

Round 2

Reviewer 2 Report

The manuscript has been significantly improved and now warrants publication in Nutrients.

Reviewer 3 Report

The quality of the current manuscript has been improved.